# Genome Resolved Biogeography of Mamiellales

**DOI:** 10.3390/genes11010066

**Published:** 2020-01-07

**Authors:** Jade Leconte, L. Felipe Benites, Thomas Vannier, Patrick Wincker, Gwenael Piganeau, Olivier Jaillon

**Affiliations:** 1Génomique Métabolique, Genoscope, Institut de Biologie François Jacob, Commissariat à l′Énergie Atomique (CEA), CNRS, Université Évry, Université Paris-Saclay, 91057 Évry, France; jleconte@genoscope.cns.fr (J.L.); thomas.VANNIER@univ-amu.fr (T.V.); pwincker@genoscope.cns.fr (P.W.); 2Research Federation for the Study of Global Ocean Systems Ecology and Evolution, FR2022/Tara Oceans GOSEE, 3 rue Michel-Ange, 75016 Paris, France; 3Observatoire Océanologique, UMR 7232 Biologie Intégrative des Organismes Marins BIOM, CNRS, Sorbonne Université, F-66650 Banyuls-sur-Mer, France; benites@obs-banyuls.fr

**Keywords:** Mamiellales, biogeography, *Tara* Oceans, sexual reproduction, mating-type, ecogenomics

## Abstract

Among marine phytoplankton, Mamiellales encompass several species from the genera *Micromonas*, *Ostreococcus* and *Bathycoccus*, which are important contributors to primary production. Previous studies based on single gene markers described their wide geographical distribution but led to discussion because of the uneven taxonomic resolution of the method. Here, we leverage genome sequences for six Mamiellales species, two from each genus *Micromonas*, *Ostreococcus* and *Bathycoccus,* to investigate their distribution across 133 stations sampled during the *Tara* Oceans expedition. Our study confirms the cosmopolitan distribution of Mamiellales and further suggests non-random distribution of species, with two triplets of co-occurring genomes associated with different temperatures: *Ostreococcus lucimarinus*, *Bathycoccus prasinos* and *Micromonas pusilla* were found in colder waters, whereas *Ostreococcus* spp. RCC809, *Bathycoccus* spp. TOSAG39-1 and *Micromonas commoda* were more abundant in warmer conditions. We also report the distribution of the two candidate mating-types of *Ostreococcus* for which the frequency of sexual reproduction was previously assumed to be very low. Indeed, both mating types were systematically detected together in agreement with either frequent sexual reproduction or the high prevalence of a diploid stage. Altogether, these analyses provide novel insights into Mamiellales’ biogeography and raise novel testable hypotheses about their life cycle and ecology.

## 1. Introduction

Mamiellales is an order of green algae (Chlorophyta) that contains some of the most ecologically important groups of photosynthetic picoeukaryotes in the marine environment [1,2,3]. They are prevalent and abundant in coastal surface waters and throughout the oceanic euphotic zone [4,5,6,7] where populations can reach high densities up to 10^3^–10^5^ cells per ml [8].

This order is composed of two families. The Bathycoccaceae are represented by the genera *Ostreococcus* and *Bathycoccus*, which are distributed across a range of marine environments [2,3,9]. Within Bathycoccaceae, previous metabarcoding analyses based on regions of the highly conserved 18S ribotype suggested niche differentiation: sequences of *Ostreococcus tauri* are detected in coastal and lagoonal environments while *Ostreococcus lucimarinus* and *Ostreococcus* spp. RCC809 are more broadly distributed in oceanic open regions [6,9]. Recently, a novel *Ostreococcus* clade E group, identified by a different 18S sequence, was found to be the most prevalent Mamiellales in the Mediterranean Sea and in coastal warm temperate sites on both sides of the Atlantic [9]. The genus *Bathycoccus* is composed of two cryptic species [10,11] with identical 18s rRNA sequences but marked differences in their ITS (Internal Transcribed Spacer) region and highly divergent genome sequences. Indeed, orthologous proteins share 78% amino-acid identity and only 26 highly conserved genes (>99%) [7,12] are found. This is similar to genome divergence reported for different species [13].

The second family of the Mamiellales order is Mamiellaceae, comprising the genera *Micromonas*, *Mantoniella* and *Mamiella* that are widespread from tropical to polar regions. Within Mamiellaceae, the genus *Micromonas* comprises the most described species with a global distribution in coastal and open ocean areas and with species adapted to polar environments [12,14,15]. *Micromonas pusilla* [16] was recently split into four species, namely, *Micromonas bravo*, *Micromonas commoda*, *Micromonas polaris*, *Micromonas pusilla*, and two clades described as candidate species 1 (clade B_4) and candidate species 2 (clade B warm) [9,15].

There are many complete genomes available for Mamiellales [17]; 17 genomes have been sequenced from four *Ostreococcus* species [18,19,20], while there are two sequenced genomes for *Micromonas*, *M. pusilla* CCMP1545 isolated in 1950 in the North Atlantic sea near Plymouth (England) and *M. commoda* RCC299 isolated in the South Pacific in 1998 [5]. *Bathycoccus* has two reference genomes, one complete genome from the strain RCC1105 [10], isolated in the Banyuls bay, and a 64% complete single amplified genome co-assembly TOSAG39–1 from the Arabian sea [11]. One common feature shared by all Mamiellales genomes is the presence of two unusual “outlier” chromosomes which present striking structural and compositional differences from standard chromosomes [21]. In *O. tauri*, these chromosomes are chromosome 2 (big outlier chromosome or BOC) and chromosome 19 (small outlier chromosome or SOC) [22,23]. In *M. pusilla* CCMP1545 BOC and SOC are represented respectively by chromosome 2 and chromosome 19 while in *M. commoda* RCC299 BOC is chromosome 1 [24] and SOC is chromosome 17. In *Bathycoccus*, the outlier chromosomes are chromosome 14 (BOC) and 19 (SOC) [10]. In all of these genomes, the BOC contains one contiguous low GC content region, flanked by two high GC content regions. The low-GC region in *O. tauri* encodes two highly divergent haplotypes, and thorough phylogenetic analysis of these two haplotypes suggests that this low-GC region designates the mating-type in Mamiellales [10,25]. Gene sequences from the two mating-types have been recently described in two additional *Ostreococcus* species, *O. mediterraneus* and *O. lucimarinus*, confirming an ancient origin of this mating type in this genus [26].

Biogeography of marine organisms faces numerous technical issues owing to the complexity of the ecosystem, especially for microbes. Consequently, knowledge about species distributions has long remained very scarce and limited to visible organisms [27]. Nonetheless, more recent studies based on microscopy or DNA sequencing addressed planktonic organisms, including viruses [28,29], bacteria, unicellular eukaryotes [30,31] and small animals [32]. Most environmental genomics surveys of organisms are based on sequencing of taxonomic marker genes, known as the barcode approach. This approach relies on the specificity of the marker and on the availability of a large taxonomic reference database such as PR2 [33]. It has been demonstrated that commonly used taxonomic marker genes have uneven levels of resolution [34] and only genome sequencing can solve this issue [11]. In that context, metagenomes can be used as resources to discover and quantify the presence of organisms for which other genomic information is available. Single cell genomes or metagenome-assembled genomes are very valuable approaches in that perspective especially for uncultured organisms [35,36,37]. However, these techniques still in their infancy are biased, and while eukaryotes for which compact and abundant genomes are most likely to be successfully detected, special attention must be paid to the possibility of assembling chimeric genomes [35].

Previous analysis of *Tara* Oceans metagenomes from 122 stations using the two *Bathycoccus* genomes to recruit reads revealed that the two species rarely co-occur and occupy distinct oceanic niches with respect to depth [11]. Here, we leverage available whole genome data in Mamiellales and metagenomic sequences in 133 samples collected from 80 sites from the *Tara* Oceans expedition to investigate the biogeography of Mamiellales and the level of diversity of the putative mating-type chromosome on a global scale.

## 2. Materials and Methods

### 2.1. Genomic Resources

Six Mamiellales reference genomes including *Bathycoccus prasinos* RCC1105 and *Bathyococcus* TOSAG39–1, *M. commoda* RCC299 and *M. pusilla* CCMP1545, and *O.* RCC809 and *O. lucimarinus* strain CCE9901 (Table 1), were used to search a large number of open ocean metagenomic samples from the *Tara* Oceans expedition (Appendix A) [38]. Genomes from *O. tauri* RCC4221 and *O. mediterraneus* RCC2590 were also searched but were not found in the *Tara* metagenomic dataset. We also used genes positions data from the Pico-PLAZA platform for each genome [17]. This study focused on samples from the 0.8–5 µm organism-size fraction, corresponding to the cell size of Mamiellales, adding up to a total of 79 surface samples as well as 54 samples from the deep-chlorophyll maximum (DCM). Those 133 samples were taken from 80 different sites located in the Mediterranean Sea, Indian, Pacific and Atlantic Oceans.

Genomic abundance of each genome in those *Tara* Oceans stations was estimated by mapping metagenomic reads onto the reference genome sequences using Bowtie2 2.1.0 aligner with default parameters [39]. We filtered out alignments corresponding to low complexity regions using the dust algorithm [40] and alignments with less than 95% mean identity or with less than 30% of high complexity bases were also discarded. These identity thresholds were compatible with intraspecific levels of diversity around 1% estimated from a population genomics study in *O. tauri* [19] so that all reads recruited to a reference genome could be assumed to belong to the same species, despite intraspecific variation. We then computed relative abundances as the number of reads mapped onto genes normalized by the total number of reads sequenced for each sample. In order to avoid non-specific mapping signals, we defined a set of outlier genes for each genome. Genes with atypical mapping behavior based on the distribution of deviant numbers of recruited metagenomic reads, and organellar genes, were discarded from the analysis similarly to Seeleuthner et al. [35]. Biogeographical maps were plotted using R-packages ggplot2_2.2.1, scales_0.4.1, maps_3.1.1 and ggtree_1.6.11. We computed principal component analysis (PCA) using the vegan_2.4-1 R-package and we used the Spearman correlation coefficient to estimate correlations between relative abundances of species.

### 2.2. Environmental Analysis

To study the link between geographical distribution of species and abiotic factors, we used the physicochemical parameter values related to the *Tara* Oceans expedition sampling sites available in the PANGAEA database [41], (http://doi.org/10.1594/PANGAEA.875575, https://doi.org/10.1594/PANGAEA.875576).

We extracted the median values for a set of parameters available for each sampling location, including depth, temperature, oxygen, salinity, photosynthetically active radiation (PAR) on the sampling week using a diffuse attenuation coefficient, concentration of nitrates, nitrates + nitrites, phosphate, silicate and chlorophyll *a*. We supplemented this dataset with simulated values for iron and ammonium (using the MITgcm Darwin model [42]).

We tested whether the occurrence of the six Mamiellales in *Tara* Oceans samples was correlated with local physicochemical conditions. For each parameter, we performed a Kruskal–Wallis test with the R base stats package followed by a post-hoc Tukey’s test using nparcomp_2.6 for significant parameters (*p*-value < 0.05).

### 2.3. Mating Types

We screened the *Tara* Oceans metagenome datasets for the presence of sequences of the recently identified 23 core gene families (GFs) of the two mating types (MTs) of *O. lucimarinus* [26,43] to estimate their ratios among samples.

The two different mating types (*MT+* and *MT−*) were previously sequenced in the strains BCC118000 (MMETSP0939) and CCE9901 respectively. The latter also corresponded to the reference genome of *O. lucimarinus* used in our biogeography analysis. We mapped metagenomic reads from eleven metagenomes where the relative abundance of *O. lucimarinus* was above 0.1% on the 23 GFs (in total 16 *MT+* and 41 *MT−* genes sequences) with the same tools and thresholds as described for read recruitment on the complete genomes.

For each sample, we normalized the estimated relative abundance of *MT+* and *MT−* sequences by dividing the number of recruited metagenomic reads by the total number of reads of the corresponding sample. Secondly, to get a single value of relative abundance for *MT+* and *MT−*, we averaged the relative abundances of their corresponding genes.

Then, ratios were computed by using these mean relative gene abundances. This approach directly using mating type sequences will be referred from here as the MT genes method.

A second method to estimate the *MT+* and *MT−* ratios was based on the proportion of metagenomic reads that were mapped on the mating type locus of chromosome 2 (BOC) following the protocol described above.

The vertical coverage of metagenomic reads (number of metagenomic reads cumulatively mapped at a given position) mapped on BOC chromosome was heterogeneous, the genomic region containing the candidate mating type locus, known to have low GC composition, recruited fewer metagenomic reads than the rest of the chromosome (Appendix A). Considering that the reference genome was *MT−*, this pattern suggested that lack of coverage in this region would correspond to the presence of the opposite mating-type in the sample.

We thus estimated the relative read coverages in the low-GC genomic region of chromosome 2, which encodes the putative mating types in Mamielles [19], compared to the standard coverage of this chromosome which corresponds to both mating-types cumulated. Those values were used as a proxy for the relative proportion of the *MT−* strains in the corresponding samples. This will be referred to as the whole chromosome method.

## 3. Results

### 3.1. Genome Resolved Distribution of Mamiellales

In order to gain a global view of their geographical distribution, we estimated the proportions of metagenomic reads from each of 133 *Tara* Oceans metagenomes that were recruited to six Mamiellales genomes (Figure 1). Mamiellales were found in 68 out of the 133 samples and were distributed across all ocean regions. Altogether, the six genomes recruited 1.38% of total metagenomic reads of the merged 68 metagenomes, with a maximum of 4.8% in one sample. *Bathycoccus* TOSAG39-1 was found in 43 out of 68 different samples, making it the most cosmopolitan species, followed by *O.* RCC809, *B. prasinos*, and *M. commoda* (Table 2). The presence of *O. lucimarinus* and *M. pusilla* was detected in 11 and five samples, respectively. However, despite those relatively lower numbers, the sites where they were detected were located in different oceans, consistent with a wide geographical distribution. Taking the proportion of recruited reads to each genome as a proxy for relative species abundance [11,35], the frequencies of the six Mamiellales appeared to be similar, ranging from 0.11% to 0.65% of the total amount of metagenomic reads, with a local maximum represented by *O.* RCC809 that recruited up to 4% of metagenomic reads in a DCM sample of a station situated in the Indian Ocean. The six Mamiellales together recruited 1.4% of all metagenomic reads from 68 stations, but with a maximum of 4.8% of metagenomic reads in any given station.

All species were present in both surface and DCM samples, but three species displayed preference for one water depth. *B. prasinos* and *M. pusilla* were mostly found in surface waters while *B.* TOSAG39-1 was more frequent in DCM waters. The three other species were equally distributed between the two depths. Comparing the basins for which *Tara* Oceans samples were available, surface water from the Eastern Pacific Ocean appeared to be the only area where Mamiellales were not detected at a significant level (no species reaching 0.1% relative abundance).

A visual inspection of the geographical distribution of the six species suggested a pattern of co-occurrence between some of them. For example, in the Indian Ocean, *B.* TOSAG39-1 and *M. commoda* always appeared together with the former being dominant. To address the question of co-occurrences, we computed pairwise correlation tests between all pairs of the six Mamiellales based on their respective metagenomic abundances (Table 3). We obtained statistically significant positive correlations between *B.* TOSAG39-1, *M. commoda* and *O.* RCC809 abundances. Also, abundances of *O. lucimarinus* and *B. prasinos* were strongly correlated. *M. pusilla* being the least abundant was found in few sites but was detected at the same locations as *O. lucimarinus*. These two groups of triplets appeared geographically segregated. Finally, we compared the distributions of the six Mamiellales through a principal component analysis (PCA) based on their metagenomic abundances (Figure 2).

The first two axes of the PCA explain more than half the variance, meaning a good representation of our sample distribution on those components. This analysis showed a very clear pattern of segregation between these two triplets of genomes that have co-occurring abundances. Each triplet was composed of one of the *Bathycoccus*, *Micromonas* and *Ostreococcus*, and the two groups, when they presented high abundances, were found in distinct samples.

### 3.2. Environmental Variables Linked to Strain Presence

Adding a color corresponding to water temperature for the samples in the PCA figure clearly revealed that the two triplets were found at different ranges of temperatures, with one group found in warmer water than the other. Therefore, we analyzed the ecological preferences of the six species and statistically tested whether available physicochemical parameters were able to discriminate them (Figure 3).

From 12 physicochemical environmental parameters, a Kruskal–Wallis test was significant for four of them and temperature was the most significant. As expected, oxygen which is well-known to be directly anti-correlated to water temperature was significant. The third discriminating parameter was chlorophyll *a*, with a more significant *p*-value than nitrate and nitrite nutrients (phosphate was not significant). For each environmental parameter, we computed pairwise species statistical comparisons (post-hoc Tukey test) in order to determine whether or not genomes were found in similar environments. This test confirmed different ecological preferences in all paired species from distinct triplets (Appendix A).

### 3.3. Mating Types

We investigated the biogeography of the two candidate mating types of *O. lucimarinus* by two methods. The first method, the MT whole chromosome method, was based on differential numbers of metagenomic reads that were recruited to mating type (MT) versus non-MT loci. The second method, the MT genes method, was based on the differential numbers of metagenomic reads that could be mapped on *MT+* and *MT−* gene sequences (Methods). The two approaches provided highly correlated estimations of mating type ratios (linear regression *r^2^* = 0.99, Appendix A). However, the estimations based on *MT* genes suggested higher frequencies of the *MT+* as compared to the estimations inferred from the whole chromosome method. This may be due to the number of marker genes in the *MT+* gene set which may not be representative of the whole mating type region. Strikingly, we detected the presence of both mating types in each sample where the genome had been found. Moreover, the mating type ratio was estimated to be variable between sites but systematically biased toward the *MT+* type from 66% up to 97% (Figure 4) using the *MT* genes method. This bias was estimated to be lower, within the 52.3% to 95.2% range, when calculated from the whole chromosome. Using the latter approach, the mating type ratio was close to 50% in both surface and DCM samples of the *Tara* Oceans station 81 (52.3% and 55.0% respectively) which is the most austral site where *O. lucimarinus* was detected. These analyses provide strong evidence of the presence of both mating-types in the samples we analyzed. In addition, the mating type genes were mapped at 99.3% and 99.6% of identity on average on the *MT+* and *MT−* genes respectively, versus 98.6% on the genes located on standard regions of the genome, consistent with their very high conservation level and the lack of putative technical bias due to sequence divergence. Finally, mating type genes were only detected in samples presenting a minimum of 0.1% relative abundance of *O. lucimarinus*. The sum of *MT+* and *MT−* abundances fits the distribution of the total abundance estimated from whole genome read recruitment (Appendix A). From a technical point of view, this comparison also provides evidence that the mating type ratio can be estimated based on differential metagenomic read coverage for any species for which the mating-type region has been identified, even if the sequence of the alternative mating-type or mating-types is not yet available. In Mamiellales, the species with only one sequenced mating-type were *O.* RCC809 [43] and *B. prasinos.*

## 4. Discussion

Following a previous whole-genome-resolved biogeographic approach based on *Tara* Oceans data and focused on *B. prasinos* and *B.* TOSAG39-1 [11], we applied the same approach to four Mamiellales reference genomes from the *Micromonas* and *Ostreococcus* genus and added 11 metagenomes to the previous dataset.

This whole genome resolved biogeographic approach enabled a better taxonomic resolution than the ribotype metabarcoding approach [14,44,45], whose resolution power is limited by the very high conservation of the 18S rDNA sequence [34]. Indeed, the V9 region of the 18S sequences is identical in all species within *Ostreococcus* and *Bathycoccus* [45].

Furthermore, screening the *Tara* Oceans metabarcoding dataset [46,47] with a similar detection threshold (minimum 0.1% relative abundance) we did not detect Mamiellales in more samples as compared with the whole genome approach. Indeed, while the whole genome approach detected Mamiellales in 68 out of our 133 samples (51%), the V9 approach detected them in 70 out of 164 available samples (48%) or in 65 out of the 131 samples (50%) that are common between the two datasets. This suggests that the metabarcoding approach did not provide a better geographical resolution than the metagenomic one, probably because we were focusing here on abundant species with relatively small genomes.

The whole genome approach revealed intriguing co-occurrence patterns between triplets of species from the three different genera: *B. prasinos*, *O. lucimarinus* and *M. pusilla* or *B.* TOSAG39-1, *O.* spp. RCC809 and *M. commoda*. It remains to be shown whether these co-occurrences are the consequence of random sampling from the same water masses, as for example within the Indian Ocean or within the Gulf Stream in which we can notice similar patterns along currents, or whether these species are adapted to the same environmental conditions. A test of the neutral hypothesis could be performed by a competition experiment between these six strains at 15 and 22 °C, the mean temperature in which each triplet was detected, and the monitoring of species frequencies over time.

The paucity of these Mamiellales genomes in samples from tropical and sub-tropical Pacific Ocean contrasts with their very broad distribution in other sampled basins. Only *B.* TOSAG39-1 and *O.* RCC809 were detected in tropical and sub-tropical basins and at lower abundance than in most other basins. The oligotrophic conditions of these Pacific areas might explain this pattern, especially the very low concentration in dissolved iron, previously described as the probable factor leading to peculiar phototrophic communities and/or local adaptations [48]. Ferredoxin is an important electron transfer enzyme for photosynthesis but particularly sensitive to iron limitation. This enzyme has been described as lost in the evolution of many lineages where its function is replaced by a stress-resistant isofunctional carrier, flavin-containing flavodoxin [49]. This hypothesis must be confirmed by the sequencing of local strains, but accordingly, we noticed the presence of flavodoxin and did not find ferredoxin in the reference genomes of *O.* RCC809 and *B.* TOSAG39-1. It would also be possible to perform tests in culture to estimate the minimal concentrations of iron and other oligo-nutrients among species and strains. Quantifying the relative abundance of a genome in a natural environment by leveraging metagenomes requires specificity to avoid over- or underestimates. Overestimates could be due to the presence in the same environment of related evolutionary species containing genomic regions conserved between them, leading to an over-recruitment of metagenomic reads. Thus, a preliminary detection of specific genomic regions is necessary. As described in Seeleuthner et al. [35], selecting regions with homogeneous coverage of recruited reads helps to exclude genomic regions that are conserved between genomes present in the same environment. Underestimates could be due to multiple situations such as less abundant species close to the detection threshold and/or a very large genome or a genome for which the reference is in the form of a low-quality assembly. Low quality reference genomes could also lead to an erroneous signal in the case of a chimeric assembly. For these reasons, genomic biogeography is well adapted to the genomes of bacteria and compact eukaryotes such as the Mamiellales. In the near future, this approach will be a key and routine methodology when advances in sequencing, in particular for metagenomes, will allow the detection or even reconstitution of complex and scarce genomes at very low cost.

The recent identification of the sequences of both highly divergent mating types in *Ostreococcus* [26] enabled an additional interpretation of the genome-resolved approach in *O. lucimarinus*. Comparison of the coverage of the mating-type sequences allowed us to infer not only the presence or absence, but also the frequency of the two mating types in the natural environment. Indirect estimation of the minimum frequency of sexual reproduction in a natural population of *O. tauri* has been estimated to be very low (one meiosis every 100,000 mitoses) [19]. This low prevalence of sexual reproduction might be the consequence of the low probability of contact between non-motile cells from the two different mating types in the environment. The indirect estimation of sexual reproduction in natural populations of *O. lucimarinus* is yet unknown, but the systematic co-occurrence of strains from both mating types suggests that mating does happen. Under clonal reproduction, haploid *Ostreococcus* strains display up to 40% differences in growth rates [19] in different temperature conditions. As a consequence, sustained clonal reproduction for a significant number of generations is expected to distort mating type frequencies in any given environment. Over many different environments and taking population bottlenecks into account we may thus expect a discrete pattern of presence or absence of each mating-type. Information about mating-type frequency in the natural environment is scarce, and the sampling and sequencing of 12 *O. tauri* strains revealed that the 12 strains all belonged to the *MT+* mating type [19], and more recently the sequencing of four *O.* spp. RCC809 single amplified genomes from the Indian Ocean also all encoded the *MT+* type [43]. In our study, the mating type ratio seems to be slightly unbalanced with a bias towards a higher frequency of the *MT+* mating type of *O. lucimarinus* in all 11 sites, while the ratio was close to 50/50 in one station. This suggests a tendency towards a higher prevalence of *MT+* types in the natural environment. Heterogametic diploids have been observed in other green algae in which sexual reproduction is well studied [50]. Diploid *Ostreococcus* zygotes are therefore expected to be heterogametic, that is containing one *MT+* and one *MT−* chromosome. However, frequent phases of clonal reproduction of haploid strains with different growth rates [19] are poised to distort mating type ratios. To our knowledge, no diploid Mamiellales has yet been observed and maintained in culture in the lab. This is not surprising since it is yet unknown how to induce meiosis in this group, and because the lab culture conditions have been optimized to sustain haploid microalgal growth. The identification of mating-types in additional species opens many novel research opportunities into the life cycle of these microalgae. Practically, knowledge about the frequency of the two mating types in the natural environment indicates how to scale up isolation protocols to increase the probability of finding both mating types. The hypothesis of a different fitness of the two mating types, suggestive of a cryptic anisogamy, could be tested experimentally for species where both *MT* have been maintained in culture.

In conclusion, the sequencing of additional metagenomes and reference genomes will improve this type of genome-based approach to a wide range of environments in many yet understudied lineages of ecologically relevant phytoplanktonic microalgae, and beyond. This is poised to reveal unprecedented insights into their ecology, i.e., interactions with other species, as well as insights into the genomic basis of their evolution and haploid–diploid life cycle alternation.

## Figures and Tables

**Figure 1 genes-11-00066-f001:**
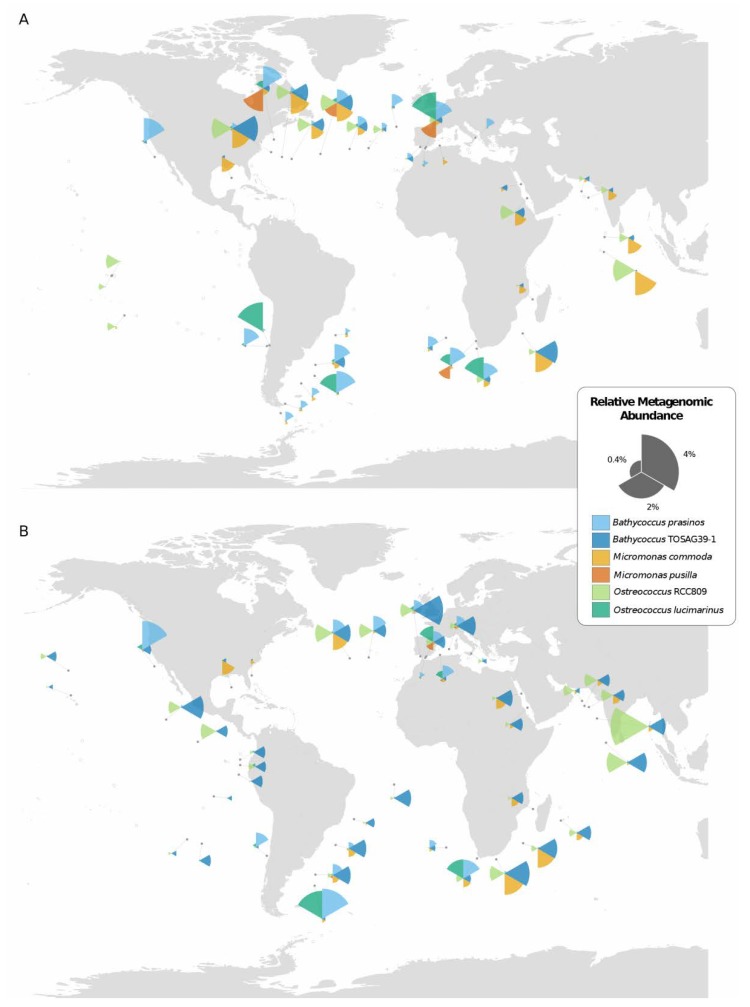
Geographical distribution of the six Mamiellales genomes in *Tara* Oceans stations from (**A**) surface and (**B**) deep-chlorophyll maximum (DCM) waters, as inferred from the relative abundance of recruited metagenomic reads. Samples with less than 0.1% relative abundance of a species are displayed as an empty circle. The sizes of the segments of coxcomb charts indicate the relative genomic abundances of the corresponding Mamiellales.

**Figure 2 genes-11-00066-f002:**
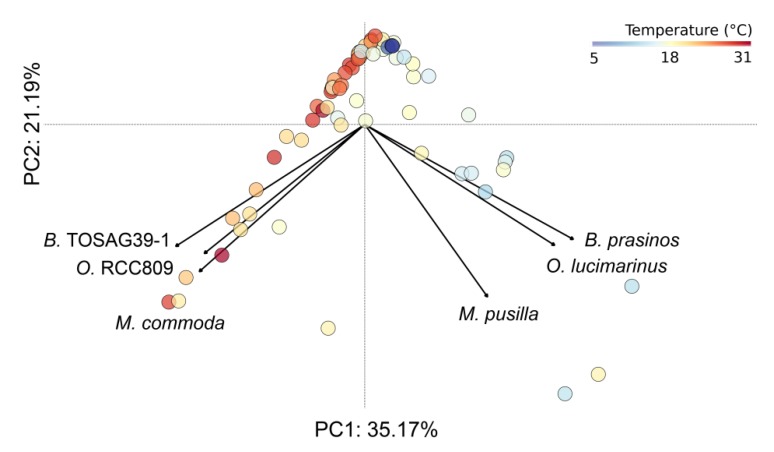
Principal component analysis computed on relative metagenomic abundances of the six Mamiellales. Each circle corresponds to a sample and is colored according to water temperature.

**Figure 3 genes-11-00066-f003:**
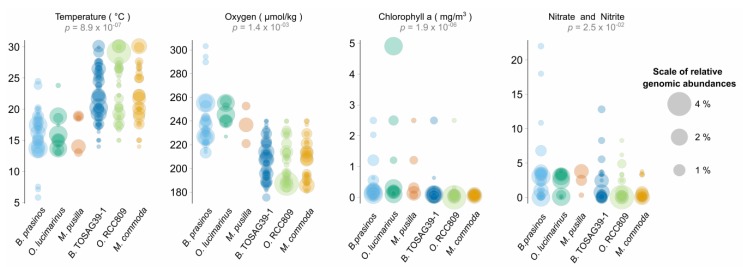
Ranges of values of environmental parameters where a significant difference was detected among Mamiellales species. Each circle corresponds to the relative metagenomic abundance of corresponding species in a given sample. A sample where several species are present is thus represented for these corresponding species (at the same value on the Y axis) but possibly with different circle sizes. *p*-values correspond to a Kruskal–Wallis test using the six Mamiellales (non-significant environmental parameters are not shown).

**Figure 4 genes-11-00066-f004:**
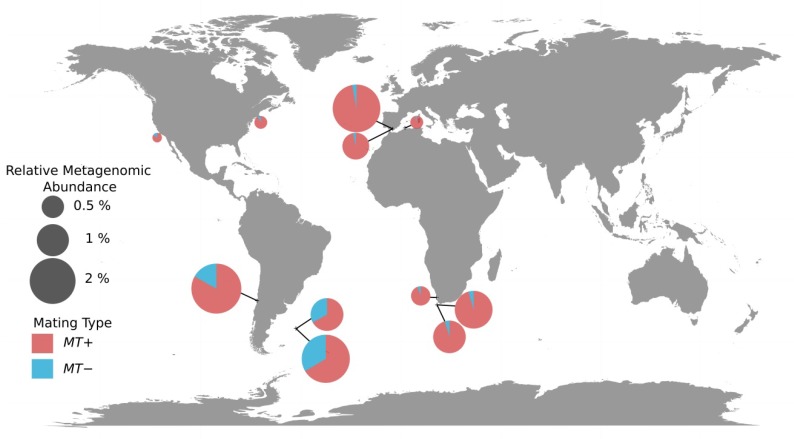
Geographical distribution of the two mating types of *Ostreococcus lucimarinus* in *Tara* Oceans stations. Each pie-chart represents a sample with a size relative to the relative metagenomic abundance of whole genome, and shows the average proportions of the two mating types genes. Ratios were determined using the MT genes method (Methods).

**Table 1 genes-11-00066-t001:** Genome data used.

Species	Source	Genome Size	Sampling Site	Sampling Year
*B. prasinos* (RCC1105)	pico-PLAZA [17]	15.1 Mb	Mediterranean Sea, France, Banyuls Bay	2006
*B.* spp. TOSAG39-1	*Tara* Oceans single-cell [11]	10.3 Mb (64% complete)	Indian Ocean, Arabian Sea, Station TARA_039	2010
*M. pusilla* (CCMP1545)	pico-PLAZA [17]	21.9 Mb	Atlantic Ocean, United Kingdom, near Plymouth	1950
*M. commode* (RCC299)	pico-PLAZA [17]	20.9 Mb	Pacific Ocean, Equatorial Pacific, New Caledonia	1998
*O. lucimarinus* (CCE9901)	pico-PLAZA [17]	13.2 Mb	Pacific Ocean, USA, California	2001
*O.* spp. RCC809	pico-PLAZA [17]	13.3 Mb	Atlantic Ocean, Tropical Atlantic, international waters	1991

**Table 2 genes-11-00066-t002:** Mamiellales repartition and abundances among *Tara* Oceans samples.

Species	Number of Samples (Abundance > 0.1%)	Percentage of Reads (Merged Samples with Reference)	Maximum Abundance in Sample
*B. prasinos*	33	0.65%	2.54%
*B.* TOSAG39-1	43	0.54%	2.41%
*M. commoda*	27	0.43%	1.70%
*M. pusilla*	5	0.11%	1.47%
*O.* RCC809	34	0.54%	4.07%
*O. lucimarinus*	11	0.35%	2.36%
Total	68	1.38%	4.80%

**Table 3 genes-11-00066-t003:** Correlations between occurrences of the six Mamiellales genomes.

Species	*B. prasinos*	*O. lucimarinus*	*M. pusilla*	*B.* TOSAG39-1	*O.* RCC809
***M. commoda***	−0.20	−0.14	−0.01	0.37 ***	0.28 *
***O*. RCC809**	−0.19	−0.13	−0.05	0.33 ***	
***B.* TOSAG39-1**	−0.22	−0.21	−0.09		
***M. pusilla***	0.31	0.24 *			
***O. lucimarinus***	0.47 ***				

Level of confidence: * *p*-value < 0.05, *** *p*-value < 0.001.

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
