# Peer review of "Genome Resolved Biogeography of Mamiellales"

_genes, 2020, doi:10.3390/genes11010066_

Round 1

Reviewer 1 Report

The authors used the data from TARA Ocean expedition, which provided a lot of valuable information about genetic diversity of single-cell eukaryotes. The group Mamiellales (Chlorophyta) is known to be abundant and broadly dispersed in nature. Thus, it is a good object for the biogeographical study. Moreover, photosynthetic picoplankton plays a significant role in water ecosystems and with climate changing this role becomes more and more prominent. The authors tried to connect the data of reference genomes and metagenomic data, based on environmental DNA. This is very important in case of single-cell eukaryotes: most of their lineages are not cultivated and the one option is to study them in nature. Unlike DNA metabarcoding studies, metagenomics allows to compare not only the frequency of the ribosomal gene, but other interesting genes like sequences of different mating type. In near future studies similar to this one should become more and more popular.

I think that the ms is of broad interest and most of its parts are well written. Some general comments:

Introduction

- Now it is focused only on the Mamiellales. Please, add some information about using genomic data in biogeographic (phylogeographic) studies, especially those concerning single-cell eukaryotes. What are they, how successful, what information (beside just novel diversity) provided.

- 37-39 line: I suggest to write all about Bathycoccaceae at first and then start to write what genera are included in Mamiellaceae and about this family.

Results

- 161-163 - “Mamiellales was not found in Eastern Pacific Ocean” - can you give any explanation of this in the Discussion? Do you think that this is a reality or some problem with sampling?

- 212-224 – I suggest to move most of the paragraph in Methods. Keep only obtained facts in the Results.

Discussion

It is quite short now, the results may be discussed a bit more.

- As there are not many studies using genomic in protist global biogeography, it will be usefull if you add a few more phrases about the method: potential limitations, problems, which could occur during the work with metagenomic data and your success in avoiding them, etc.

- 278 - “some kind of mutualistic interactions” - can you write about any examples, when free-living microalgae interact with each other in such way? Interesting, that you found members of all three genera in each of the triplet. Perhaps, this indeed shows how two populations of Bathycoccus sp. (/Micromonas/Ostreococcus) separated in past in two different species. To test the strains at different temperatures is a good idea, hopefully, someone will do that. Phylogenomic study is also desired (after obtaining more Mamiellales genomes).

- 281 – 310 – the paragraph could be written more clearly. Now for me it is hard to understand, what is based on your results and what is known from literature.

Author Response

Point-by-point answers are listed here after in red after each comment.

Introduction

- Now it is focused only on the Mamiellales. Please, add some information about using genomic data in biogeographic (phylogeographic) studies, especially those concerning single-cell eukaryotes. What are they, how successful, what information (beside just novel diversity) provided.

We have added to the Introduction a paragraph about marine biogeography and the impact of genomics on the study of these single-cell eukaryotes

 - 37-39 line: I suggest to write all about Bathycoccaceae at first and then start to write what genera are included in Mamiellaceae and about this family.

Done

Results

- 161-163 - “Mamiellales was not found in Eastern Pacific Ocean” - can you give any explanation of this in the Discussion? Do you think that this is a reality or some problem with sampling?

We have no evidence of any issue due to the sampling as these samples were correctly analyzed (De Vargas et al. 2015, Carradec et al. 2018, Ibarbalz et al. 2019). We added a paragraph in the discussion suggesting an impact of the particular oligotrophic conditions in the Pacific.

- 212-224 – I suggest to move most of the paragraph in Methods. Keep only obtained facts in the Results.

Almost all this paragraph has been moved as suggested and slightly rewritten to fit with Methods.

Discussion

It is quite short now, the results may be discussed a bit more.

- As there are not many studies using genomic in protist global biogeography, it will be useful if you add a few more phrases about the method: potential limitations, problems, which could occur during the work with metagenomic data and your success in avoiding them, etc.

As suggested, a paragraph was added to the discussion regarding biogeography methods and limitations.

- 278 - “some kind of mutualistic interactions” - can you write about any examples, when free-living microalgae interact with each other in such way? Interesting, that you found members of all three genera in each of the triplet. Perhaps, this indeed shows how two populations of Bathycoccus sp. (/Micromonas/Ostreococcus) separated in past in two different species. To test the strains at different temperatures is a good idea, hopefully, someone will do that. Phylogenomic study is also desired (after obtaining more Mamiellales genomes).

We agree with the reviewer that this sentence is too speculative without additional information. The sentence was deleted from the text to keep only the similar adaptation of three species hypothesis.

- 281 – 310 – the paragraph could be written more clearly. Now for me it is hard to understand, what is based on your results and what is known from literature?

We have tried to clarify the paragraph, indicating which results are from our own study

Reviewer 2 Report

Review for manuscript genes-669008 

Mamiellales comprise many ecologically important green alga species and are found in a wide range of marine habitats. Previous biogeographic studies of some species in this order were often limited by the low resolution of ribosomal markers indicating, nevertheless, distinct preferences for either coastal or open ocean regions.

The manuscript “Genome resolved Biogeography of Mamiellales” by Leconte and co-authors describes the biogeographic distribution of 6 Mamiellales species inferred from metagenomic reads mapping to assembled genomes of the respective organisms. This analysis revealed interesting distribution patterns following water temperature. The authors also studied the relative abundance of MT+ and MT- mating types of Ostreococcus lucimarinus in the metagenome samples, suggesting co-occurrence of both mating types in all samples but higher abundance of MT+.

Overall, the manuscript is well written and provides some interesting new insights. The methodological approach to identifying mating type ratios from metagenome data has great potential to be applied in other studies and contexts. However, certain details throughout the text need to be clarified (e.g. legends and labels of figures) and some typos corrected. The manuscript would in general benefit from clear nomenclature regarding the two different mating type mapping approaches. At the moment, it is difficult to distinguish them and correctly guess to which method the authors are referring. Maybe the authors could give them descriptive names in Material and Methods and refer to them in this way throughout the Figures, Results and Discussion. Furthermore, the discussion regarding the mating type ratios is confusing, as the authors refer to Ostreococcus as a diploid organism. In case the authors are referring to the diploid zygote stage, I do not understand how it can significantly affect the mating type ratio when it occurs to rarely. The rationale behind this paragraph (line 299-304) needs to be better explained. Further, more detailed remarks are listed below.  

Abstract

Line 15: single gene markers 

Introduction

Line 38: Ostreococcus and Bathycoccus, which are distributed…

Line 53: “regrouping” does not sound appropriate. I would suggest: “the genus Micromonas comprises most described species with a global distribution…”

Line 73: Maybe better: “this low-GC region designates the mating-type…”

Line 74: two mating types

Material and Methods

Line 90: What do you mean with “coding gene positions”?

Line 92: Please, introduce the abbreviation DCM here.

Line 103: Do you mean genomes instead of genes?

Line 105: Please, define metagenomic RPKM distribution.

Line 106: What is “World map”? A package, a software?

Line 108: correlation between relative abundance of species

Line 130: “Two mating types were available” sounds a bit awkward, as there can only be two. I would suggest: “The two different mating types (MT+ and MT-) were sequenced in the strains BCC119000 and CCE9901, respectively.”

Line 131: There’s a comma missing between “CCE9901” and “which”.

Line 133: read recruitment on the complete genomes of eleven metagenomes…

Line 136: I don’t understand this sentence.

Line 135-139: It’s very hard to understand, which approach was chosen for mapping against the reference genome CCE9901 and which approach for mapping against the individual mating type chromosome for MT-. What’s the technical differences between these approaches and why did you chose different methods (Bowtie vs. samtools mpileup)?

Results

Line 142: I think that the term “coverage” is not completely appropriate and I would instead suggest “the recruitment of reads to six Mamiellales genoms…”.

Line 143: 133 metagenomes from the Tara Oceans initiative

Line 158: Please, define the abbreviation DCM earlier!

Line 158-159: This sentence is hard to understand. I would suggest: “, but three species displayed preferences for one of the habitats.”

Line 163: What do you call a “significant level”?

Line 176: Table 3 shows significant correlation between relative abundance of M. pusilla and O. lucimarinus. It seems like the sample sized was high enough to detect this correlation.

Figure 2: The temperature legend is missing red.

Line 194: The term “preference” is not appropriate, as you would have to test it in growth experiments with the individual strains. I suggest using “niche” instead.

Line 200: , with a more significant p-value… (phosphate was not significant)

Line 201-202: I don’t understand what you mean with “ranges of values that correspond to habitats of organisms”, as you don’t compare different environmental parameters.

Line 203: I suggest reformulating this sentence to improve readability. Maybe: “This test confirmed different ecological niches in all paired species from distinct triplets.”

Figure 3: Please, specify in the caption, if all circles in a horizontal line correspond to one sample.

Line 213: gene sequence markers

Line 214: , which encode for the…

Line 215: was heterogeneous. Heterogeneous among samples or genes in the low-GC region?

Line 216: It recruited less metagenomic reads than what?

Figure S1: What is Figure S2 showing? The entire low GC region or the BOC? Do the low coverage 300-700bp correspond to the MT genes?

Line 226: To which protocol are you referring? A previous study or a previous part of this manuscript?

Figure S2: The labels of the axes are confusing, as BOC hasn’t been mentioned in the manuscript since the introduction. Please, try to find a more intuitive nomenclature that facilitates distinguishing between the two approaches to estimate relative mating type abundance.

Line 242-245: Please, break up this sentence. It’s too long. Line 244 also reads awkwardly. I suggest: “The sum of MT+ and MT- abundances fitted the distribution of the total abundance estimated from whole genome read recruitment.”

Figure 4: It is not clear to me, on which approach to estimate mating type abundance this figure is based. A consistent, intuitive nomenclature for the two methods would be helpful.

Discussion

Line 256: whole-genome-resolved biogeographic approach

Line 263: What do you mean with “amalgamates”? I don’t understand this term.

Line 267: in instead of win

Line 274-278: Please, break up this sentence. It’s too long.

Line 292: Please, provide a reference or this statement.

Line 297: “… in all 11 sites, while the ratio is close…”

Line 299-301: What do you mean with “diploid Ostreococcus”? The vegetative cells are haploid, only the zygote is diploid. To my understanding, the term “heterogametic” refers to vegetative, diploid cells in which the sex chromosomes differ. I could not find any reference to “heterogametic diploids among green algae” in [40]. To which other algal species are you referring?

Line 301-302: Please, provide a reference for this statement! Do you assume that natural selection acts against one of the mating types (MT-) and in this way clonal reproduction distorts the mating type ratio?

Line 303-304: What do you mean with “no diploid Mamiellales has yet been maintained in culture”? In most microalgal species, sexual reproduction and formation of zygotes is difficult to induce in the lab. However, many diploid microalgae such a diatoms successfully grow in culture. Do you specifically refer to sexual reproduction in Mamiellales at the end of the sentence?

Line 312: “bounty” is an unusual term. I suggest: “to a wide range of available metagenomes in many yet understudied lineages”

Author Response

Point-by-point answers are listed here after in red after each comment.

Abstract

Line 15: single gene markers Done                                                 

Introduction

Line 38: Ostreococcus and Bathycoccus, which are distributed… Done

Line 53: “regrouping” does not sound appropriate. I would suggest: “the genus Micromonas comprises most described species with a global distribution…” Done

Line 73: Maybe better: “this low-GC region designates the mating-type…” Done

Line 74: two mating types Done

Material and Methods

Line 90: What do you mean with “coding gene positions”?

We mean positions of protein coding genes in the genome. We have rephrased.

Line 92: Please, introduce the abbreviation DCM here. Done

Line 103: Do you mean genomes instead of genes?

We did not consider intergenic regions, the sentence is correct.

Line 105: Please, define metagenomic RPKM distribution.

This analysis relies on the assumption that the number of reads recruited per gene follows a normal distribution in a sample. This is described in detail in Seeleuthner et al. We now avoid this terminology.

Line 106: What is “World map”? A package, a software?

“World map” was changed for “Biogeographical maps”, as it was referring to figures. “Performed” was changed to “plotted”.

Line 108: correlation between relative abundance of species. Done

Line 130: “Two mating types were available” sounds a bit awkward, as there can only be two. I would suggest: “The two different mating types (MT+ and MT-) were sequenced in the strains BCC119000 and CCE9901, respectively.” Done

Line 131: There’s a comma missing between “CCE9901” and “which”. Sentence changed

Line 133: read recruitment on the complete genomes of eleven metagenomes… Done

Line 136: I don’t understand this sentence.

We have rephrased to better explain the methodology.

Line 135-139: It’s very hard to understand, which approach was chosen for mapping against the reference genome CCE9901 and which approach for mapping against the individual mating type chromosome for MT-. What’s the technical differences between these approaches and why did you chose different methods (Bowtie vs. samtools mpileup)?

We clarified in the text the technical differences between the two methods. One method compares the number of metagenomic reads that were recruited in the mating type and in the non-mating type loci of a MT- whole chromosome. The second method compares the number of metagenomic reads that were recruited on MT+ and MT- gene sequences.

Results

Line 142: I think that the term “coverage” is not completely appropriate and I would instead suggest “the recruitment of reads to six Mamiellales genoms…”. Done

Line 143: 133 metagenomes from the Tara Oceans initiative Done

Line 158: Please, define the abbreviation DCM earlier! Done

Line 158-159: This sentence is hard to understand. I would suggest: “, but three species displayed preferences for one of the habitats.” Done

Line 163: What do you call a “significant level”?

Added in the text: no species reaching 0.1% relative abundance

Line 176: Table 3 shows significant correlation between relative abundance of M. pusilla and O. lucimarinus. It seems like the sample sized was high enough to detect this correlation.

This is a good point. We edited the text.

Figure 2: The temperature legend is missing red. Fixed

Line 194: The term “preference” is not appropriate, as you would have to test it in growth experiments with the individual strains. I suggest using “niche” instead.

“Preferences” is not necessarily related to lab experiments and can be used as well for in-situ observations as we find some species in defined ranges of temperatures. Several publications use this term (Vannier et al. 2016)

Line 200: , with a more significant p-value… (phosphate was not significant) Done

Line 201-202: I don’t understand what you mean with “ranges of values that correspond to habitats of organisms”, as you don’t compare different environmental parameters.

Rephrased in the text. We are comparing species pairwise and thus trying to determine physico-chemical differences in their environments (such as temperature) one parameter at a time.

Line 203: I suggest reformulating this sentence to improve readability. Maybe: “This test confirmed different ecological niches in all paired species from distinct triplets.” Done

Figure 3: Please, specify in the caption, if all circles in a horizontal line correspond to one sample.

More details were added, one vertical line is a species, one circle is a sample placed on the Y-axis at the same level for the 6 species but with a size proportional to the genomic abundance.

Line 213: gene sequence markers Done

Line 214: , which encode for the… Done

Line 215: was heterogeneous. Heterogeneous among samples or genes in the low-GC region?

Added in text, it was heterogeneous between different parts of the chromosome.

Line 216: It recruited less metagenomic reads than what?

Added in text, less than the rest of the chromosome

Figure S1: What is Figure S1 showing? The entire low GC region or the BOC? Do the low coverage 300-700bp correspond to the MT genes?

Figure S1 is showing the BOC, and the low coverage regions do indeed mostly correspond to MT genes (but all genes are not necessarily MT genes)

Line 226: To which protocol are you referring? A previous study or a previous part of this manuscript?

Sentences were changed. We were referring to the biogeography method in the first part of this manuscript.

Figure S2: The labels of the axes are confusing, as BOC hasn’t been mentioned in the manuscript since the introduction. Please, try to find a more intuitive nomenclature that facilitates distinguishing between the two approaches to estimate relative mating type abundance.

A nomenclature was added; labels in the figure S2 were edited.

Line 242-245: Please, break up this sentence. It’s too long. Line 244 also reads awkwardly. I suggest: “The sum of MT+ and MT- abundances fitted the distribution of the total abundance estimated from whole genome read recruitment.” Done

Figure 4: It is not clear to me, on which approach to estimate mating type abundance this figure is based. A consistent, intuitive nomenclature for the two methods would be helpful.

Nomenclature added. This figure corresponds to the method now names “MT genes method”, as indicated in figure 4 legend.

Discussion

Line 256: whole-genome-resolved biogeographic approach Done

Line 263: What do you mean with “amalgamates”? I don’t understand this term.

Edited in text: the V9 region of the 18S sequences is identical in all species within Ostreococcus and Bathycoccus

Line 267: in instead of win Done

Line 274-278: Please, break up this sentence. It’s too long. Done

Line 292: Please, provide a reference or this statement.

We have now rephrased the sentence to clarify this point

Line 297: “… in all 11 sites, while the ratio is close…” Done

Line 299-301: What do you mean with “diploid Ostreococcus”? The vegetative cells are haploid, only the zygote is diploid. To my understanding, the term “heterogametic” refers to vegetative, diploid cells in which the sex chromosomes differ. I could not find any reference to “heterogametic diploids among green algae” in [40]. To which other algal species are you referring?

We have now clarified this and replaced diploid Ostreococcus by "diploid Ostreococcus zygote". The zygote may be expected to be heterogametic, as in the green algae Chlamydomonas and Volvox.

Line 301-302: Please, provide a reference for this statement! Do you assume that natural selection acts against one of the mating types (MT-) and in this way clonal reproduction distorts the mating type ratio?

Not necessarily, we have just previously observed that each strain -- and this applies to strains from different MT too -- have different growth rates at different T°C and salinity conditions, so that these different growth rates in the absence of sexual reproduction, will logically lead to distorted mating type ratios. We have now rephrased this:

However, frequent phases of clonal reproduction of haploid strains with different growth rates (Blanc-Mathieu et al. 2017) are poised to distort mating type ratios.

Line 303-304: What do you mean with “no diploid Mamiellales has yet been maintained in culture”? In most microalgal species, sexual reproduction and formation of zygotes is difficult to induce in the lab. However, many diploid microalgae such as diatoms successfully grow in culture. Do you specifically refer to sexual reproduction in Mamiellales at the end of the sentence?

We have now rewritten this sentence to clarify this point : To our knowledge, no diploid Mamiellales have yet been observed and maintained in culture in the lab, but this is not surprising since it is yet unknown how to induce meiosis in this group, and because the lab culture conditions have been optimized to sustain haploid microalga growth.

Line 312: “bounty” is an unusual term. I suggest: “to a wide range of available metagenomes in many yet understudied lineages” Done